# Beyond Two-Stage Training: Integrating SFT and RL for Improved Reasoning in LLMs

## Abstract

Reinforcement learning (RL) has proven effective in incentiving the reasoning abilities of large language models (LLMs), but faces significant efficiency challenges due to its extensive trial-and-error nature. A common practice is to employ supervised fine-tuning (SFT) as a warm-up stage; however, this decoupled two-stage approach limits interaction between SFT and RL, thereby constraining overall effectiveness. This study introduces a novel method for learning reasoning models that employs bilevel optimization to facilitate better cooperation between these training paradigms. Specifically, the SFT objective are explicitly conditioned on the optimal solution of the RL objective. During training, lower-level updates enable the model to receive SFT supervision concurrently with RL-based exploration, while upper-level updates are optimized to ensure that the joint training yields higher rewards than RL alone. Empirical evaluations on five reasoning benchmarks demonstrate that our method consistently outperforms baselines and achieves a better balance between effectiveness and efficiency.

## 1   Introduction

The emergence of OpenAI's o1 [21] and DeepSeek-R1 [7] represents a profound paradigm shift in Large Language Models (LLMs). Test-time scaling enables these models to execute longer Chain-of-Thought reasoning, inducing sophisticated reasoning behaviors. This capability makes them particularly effective in challenging domains such as mathematics [5, 11] and programming problems [2, 6].

The central technique driving this progress is is large-scale, rule-based reinforcement learning (RL), which induces sophisticated reasoning behaviors by exploring the reward signal. However, the inherently trial-and-error nature of RL renders the training process highly inefficient. An alternative approach is supervised fine-tuning (SFT) on curated long chain-of-thought (CoT) datasets, which enables models to rapidly acquire effective reasoning patterns through imitation learning. While more sample-efficient, SFT is typically less generalizable than RL. In practice, state-of-the-art training pipelines often adopt a two or multi-stage paradigm, using SFT as a warm-up phase before applying RL. For example, DeepSeek-R1 [7] undergoes multiple rounds of SFT and RL to refine reasoning performance. However, in these two or multi-stage pipelines, SFT and RL training are typically performed in a fully decoupled manner. This raises a natural question:

> ***Can we design a training method that enables meaningful information exchange
> between the SFT and RL paradigms?***

To investigate this, we first propose a simple baseline that alternates between SFT and RL updates during training. Despite its simplicity, this approach improves both convergence efficiency and final

performance. Building on this insight, we further develop a bilevel optimization framework, in which SFT is formulated as the upper-level problem and RL as the lower-level problem. By solving this nested optimization objective, the SFT updates are explicitly conditioned on the RL solution, allowing SFT to provide more targeted guidance to RL. This ultimately yields a model that aligns well with both supervised and reward-driven objectives.

Specifically, we implement this bilevel structure using two learnable components: a base model and a set of LoRA modules, which together form an augmented model. The base model is optimized using RL as the lower-level objective, while the LoRA parameters are updated through a supervised upper-level objective. To make this bilevel optimization tractable, we introduce a penalty-based relaxation strategy, where the relaxed upper-level update *explicitly encourages cooperation by maximizing the reward gap between joint SFT+RL training and RL-only optimization*. In doing so, the upper-level optimization shapes the lower-level dynamics, fostering tighter alignment between supervised learning and reinforcement learning, and improving overall training efficiency.

To validate the effectiveness of our approach, we conduct experiments using the Qwen-2.5 3B model trained on the LIMR dataset, a challenging mathematical reasoning benchmark constructed from MATH [10]. We evaluate performance across six diverse benchmark datasets covering both standard and competition-level tasks. Our results demonstrate consistent improvements over six strong baselines, including supervised fine-tuning, zero-shot RL, and multi-stage SFT+RL pipelines. Notably, our method achieves superior performance in terms of both accuracy and training efficiency, confirming the benefits of tightly integrating SFT and RL through bilevel optimization.

Our work makes the following three contributions:

1. **Comparative analysis of reasoning training paradigms.** We systematically analyze and compare three prevalent strategies for training reasoning-capable language models: supervised fine-tuning (SFT), reinforcement learning (RL), and multi-stage SFT+RL pipelines. Based on this analysis, we introduce a simple yet effective alternative baseline that addresses the lack of interaction in conventional two-stage training setups.

2. **A bilevel optimization framework for integrating SFT and RL.** To promote meaningful cooperation between SFT and RL, we propose a bilevel optimization method named *BRIDGE*. BRIDGE formalizes SFT as the upper-level objective and RL as the lower-level objective, and employs a penalty-based relaxation to explicitly encourage joint training to achieve higher rewards than RL alone by maximizing the reward gap between the two.

3. **Empirical validation on six mathematical reasoning benchmarks.** We conduct extensive experiments using the Qwen-2.5 3B model trained on the LIMR dataset and evaluated across six diverse reasoning benchmarks. Our method consistently outperforms strong baselines in terms of both accuracy and training efficiency, demonstrating the practical benefits of tightly integrated SFT-RL optimization.

## 2  Preliminaries

We begin by reviewing three prevalent fine-tuning strategies for training reasoning models and conduct a comparative analysis. We then introduce a simple yet effective improved baseline.

### 2.1  Fine-tuning Methods for Reasoning Models

We consider a large language model (LLM) parameterized by $\boldsymbol{\theta}$, which defines a conditional distribution $\pi(y|x; \boldsymbol{\theta})$ over output sequences $y$ given input sequences $x$. This work focuses on three widely used methodologies for tuning $\boldsymbol{\theta}$ to incentivize the model's reasoning capabilities.

**Rule-based Reinforcement Learning.** Reinforcement learning with verifiable rewards has gained increasing attention for its effectiveness in training advanced reasoning models such as DeepSeek-R1 [7]. Given a dataset $\mathcal{D}_{\mathrm{RL}} := \{(x, y)\}$ with verifiable outputs—such as mathematics competition problems or programming tasks—the objective of rule-based RL is formulated as:

$$
\begin{aligned}
\max_{\boldsymbol{\theta}}\ J_{\mathrm{RL}}(\boldsymbol{\theta}) := &\ \mathbb{E}_{(x,y)\sim\mathcal{D}_{\mathrm{RL}},\ \hat{y}\sim\pi(\cdot|x;\boldsymbol{\theta})} \left[ r(\hat{y}, y) \right] \\
& - \mathbb{E}_{(x,y)\sim\mathcal{D}_{\mathrm{RL}}} \left[ D_{\mathrm{KL}} \left( \pi(\cdot \mid x; \boldsymbol{\theta}) \,\|\, \pi_{\mathrm{ref}}(\cdot \mid x) \right) \right]
\end{aligned}
\tag{1}
$$

where $\pi_{\mathrm{ref}}$ is a fixed reference model, and $r(\hat{y}, y)$ is a rule-based reward function that evaluates the correctness of predictions using a binary signal:

$$r(\hat{y}, y) = \begin{cases} 1, & \text{if } \hat{y} \equiv y, \\ -1, & \text{otherwise} \end{cases} \qquad (2)$$

Here, $y$ denotes the ground-truth answer and $\hat{y}$ is the model's predicted output. The equivalence relation $\hat{y} \equiv y$ is typically computed by a domain-specific verifier (e.g., a symbolic math engine or code interpreter).

Since the KL divergence term in (1) is generally not directly computable, this objective is often solved using policy optimization methods such as Proximal Policy Optimization (PPO) [22] and Group Relative Policy Optimization (GRPO) [7].

**Supervised Fine-Tuning.** In supervised fine-tuning, we assume access to a curated dataset $\mathcal{D}_{\mathrm{SFT}} := \{(x, r, y)\}$ consisting of input prompts $x$, intermediate reasoning steps $r$ distilled from larger reasoning models, and final answers $y$. The training objective maximizes the log-likelihood of generating both the reasoning process and the answer:

$$\max_{\boldsymbol{\theta}} \ J_{\mathrm{SFT}}(\boldsymbol{\theta}) := \mathbb{E}_{(x,r,y) \sim \mathcal{D}_{\mathrm{SFT}}} \left[ \log \pi\left(r, y \mid x; \boldsymbol{\theta}\right) \right]. \qquad (3)$$

This approach encourages the model to not only produce correct answers but also to imitate expert reasoning steps that lead to those answers.

**Two-Stage Cold Start.** In practice, a common recipe is to use SFT as a warm-up stage before applying RL. This two-stage approach, often referred to as a "cold start" for RL. The SFT stage ensures that the model imitate expert reasoning patterns, which provides a good prior for subsequent reward-driven optimization.

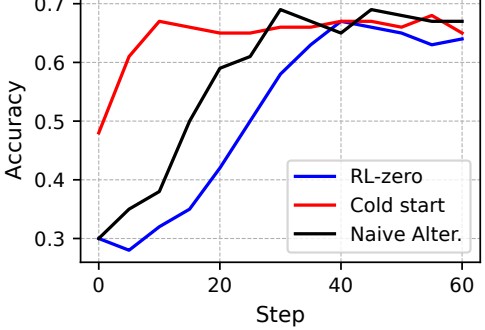

Figure 1: Comparison of Training Methods.

**Algorithm 1:** A Simple Alternating Method

1: Initialize parameters $\theta_0$; learning rates $\alpha_{\mathrm{SFT}}$, $\alpha_{\mathrm{RL}}$; datasets $D_{\mathrm{SFT}}$, $D_{\mathrm{RL}}$; total steps $T$
2: **for** $t = 1$ to $T$ **do**
3:    // RL step
4:    Sample query $x_t \sim D_{\mathrm{RL}}$
5:    Generate solution with $\pi_{\theta_{t-1}}(x_t)$
6:    Compute reward $r_t$
7:    $\theta'_{t-1} \leftarrow \theta_{t-1} + \alpha_{\mathrm{RL}} \nabla J_{\mathrm{RL}}(\theta_{t-1})$
8:    // SFT step
9:    Sample example $(x_t, y_t) \sim D_{\mathrm{SFT}}$
10:    $\theta_t \leftarrow \theta'_{t-1} + \alpha_{\mathrm{SFT}} \nabla L_{\mathrm{SFT}}(\theta'_{t-1})$
11: **end for**

## 2.2 Comparative Analysis of Fine-Tuning Strategies

We conduct a comparative study of fine-tuning strategies using the Qwen2.5-base model as the backbone. The training data consists of math problems at the grade 3–5 level, and evaluation is performed across five reasoning benchmarks, including Math500. Detailed experimental settings are provided in Section 4.1. Figure 1 illustrates how test accuracy on Math500 evolves during training.

We observe that **SFT exhibits rapid initial learning**, while **RL achieves better final convergence**. As shown in Figure 1, SFT improves accuracy quickly during the early training stages but plateaus at a suboptimal level. In contrast, RL learns more slowly but eventually surpasses SFT in final performance.

The **two-stage cold start approach combines the strengths of both paradigms**. Figure 1 further shows that the SFT warm-up phase significantly accelerates RL convergence and improves its final performance. This suggests that SFT provides a strong inductive prior, guiding the subsequent RL stage toward better optima.

These results suggest that RL and SFT offer complementary advantages in reasoning tasks, motivating further exploration of their integration.

**A Simple Alternating Baseline.** To further investigate the supportive role of SFT in reinforcement learning, we design a simple alternating optimization strategy between the two methods, as outlined in Algorithm . This approach alternates between reinforcement learning steps, which explore novel reasoning traces, and supervised fine-tuning steps, which imitate expert reasoning patterns (see Section 4.1 for details on the SFT dataset). As shown in Figure 1, this alternating strategy converges faster than pure RL and achieves better final performance than both SFT and the two-stage Cold-start approach. While this integration leads to empirical performance gains, the current formulation treats SFT and RL as independent update processes, and there is no guarantee that alternating updates consistently outperform either method alone. This limitation raises a important question: *How can we design training strategies that ensure the synergy between SFT and RL leads to guaranteed gains over standalone RL?*

# 3 Methodology

In this section, we propose BRIDGE, a framework that tightly couples SFT and RL. We will first introduce the formulation, and then the learning algorithm and explanations.

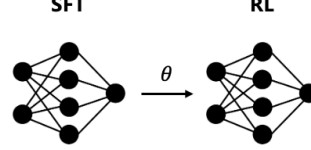

## 3.1 BRIDGE Framework

We define an augmented model $\bar{\theta} := [\theta, w]$, where $\theta$ denotes the base model parameters and $w$ represents the LoRA weights [13]. Given a long-form chain-of-thought (CoT) dataset $\mathcal{D}_{\text{SFT}}$ for supervised fine-tuning and a verifiable dataset $\mathcal{D}_{\text{RL}}$ for reinforcement learning, our objective is to integrate the supervised learning objective in Eq. (3) with the policy optimization problem in Eq. (1). To do this, we propose to solve the following bilevel optimization problem:

Figure 2: Comparison of two fine-tuning paradigms.

$$
\max_{w} \quad J_{\text{SFT}}(\theta, w) := \mathbb{E}_{(x,r,y)\sim\mathcal{D}_{\text{SFT}}}\left[\log \pi\left(r, y \mid x;\ \theta^*(w), w\right)\right]
$$

$$
\text{s.t.} \quad \theta^*(w) := \arg\max_{\theta}\left\{\mathbb{E}_{(x,y)\sim\mathcal{D}_{\text{RL}},\ \hat{y}\sim\pi(\cdot|x;\theta,w)}\left[r(\hat{y}, y)\right]\right.
$$

$$
\left.- \mathbb{E}_{(x,y)\sim\mathcal{D}_{\text{RL}}}\left[D_{\text{KL}}\left(\pi(\cdot \mid x; \theta, w)\,\|\,\pi_{\text{ref}}(\cdot \mid x)\right)\right]\right\}.
$$

$$(4)$$

The above problem has a two-level structure that draws inspiration from the leader-follower problem in game theory. SFT serves as the leader with greater decision-making power, capable of predicting the RL component's optimal response $\theta^*(w)$ for any given parameter set $w$ during training. Meanwhile, RL acts as the follower, optimizing the base model parameters $\theta$ conditional on the SFT-determined parameters $w$. During training, these two components interact dynamically to achieve enhanced cooperation, resulting in improved learning outcomes. As shown in Figure 2, this structure enables a more coordinated fine-tuning process compared to the traditional two-stage recipe.

By solving (4), we aim to find a augmented model $\bar{\theta}$ such that: if one trains the base parameter $\theta$ on $D_{RL}$, then the fine-tuned model $\theta^*(w)$ needs to fit well with the long CoT dataset $D_{SFT}$.

## 3.2 Learning Algorithm

Following the penalty-based methods [24, 26], we next consider reformulating (4) with penalty functions. Specifically, our first goal is to reformulate (4) to a closely related single-level problem that facilitates efficient gradient-based algorithms.

We define the penalty function for the sub-optimality of the lower-level problem in (4) as:

$$
p(w, \theta) = \max_{\theta'} J_{RL}(\theta', w) - J_{RL}(\theta, w) \tag{5}
$$

Given a penalty constant $\gamma \in (0, 1)$, penalizing $p(w, \theta)$ onto the upper-level objective yields the following penalized problem of (6):

$$
\max_{\theta, w}(1 - \lambda)J_{SFT}(\theta, w) - \lambda[\max_{\theta'} J_{RL}(\theta', w) - J_{RL}(\theta, w)] \tag{6}
$$

Note that in E.q. (6), the value of the term $\max_{\theta'} J_{RL}(\theta', w)$ is solely a function of $w$ and is independent of $\theta$. Then we can update $\theta$ iteratively by doing stochastic gradient ascent:

$$\theta^{k+1} = \theta^k + \alpha \left[ (1 - \lambda) \nabla_\theta J_{\text{SFT}}(\theta, w) + \lambda \nabla_\theta J_{\text{RL}}(\theta, w) \right] \tag{7}$$

The penalty strength $\gamma$ can be scheduled to increase at each epoch from a small value: in earlier epochs, we warm-start the base parameter on the long-CoT examples. Then we gradually increase $\gamma$ for increasing accuracy in solving for $\theta^*(w)$ and a solution for the original problem in (4).

To evaluate the gradient for $w$, we need to evaluate $\nabla_\omega \max'_\theta J_{RL}(\theta', w)$. We assume $J_{RL}(\theta', w)$ satisfies the conditions for Danskin's theorem, and then we can write $\nabla_\omega \max'_\theta J_{RL}(\theta', w) \approx \nabla_\omega J_{RL}(\hat{\theta}, w)$, and the above gradient approximation becomes exact if $\hat{\theta} = \theta^*(\omega)$. Given this closed-form gradient, we can update $\omega$ with the approximate stochastic gradient ascent:

$$w^{k+1} = w^k + \beta \left[ (1 - \lambda) \nabla_w J_{\text{SFT}}(\theta, w) + \lambda (\nabla_w J_{\text{RL}}(\theta, w) - \nabla_w J_{\text{RL}}(\hat{\theta}, w)) \right] \tag{8}$$

Where $\hat{\theta}$ is the approximation of $\theta^*(\omega)$ obtained by taking one gradient ascend step on $\theta$ with respect to the $J_{RL}$ objective:

$$\hat{\theta}_{k+1} = \hat{\theta}_k - \alpha \nabla_\theta J_{RL}(\hat{\theta}_k, w) \tag{9}$$

---

**Algorithm 2:** Learning Algorithm of BRIDGE

1: Initialize augmented parameters $\bar{\theta}^0 = (\theta^0, w^0)$, and auxiliary parameters $\hat{\theta}^0 := \theta^0$;
 learning rates $\alpha$, $\beta$; penalty weight $\lambda$; number of iterations $K$
2: **for** $k = 0$ to $K - 1$ **do**
3:     Sample mini-batches $D_{\text{SFT}}$ and $D_{\text{RL}}$.
4:     Compute supervised objective $J_{\text{SFT}}(\theta^k, w^k)$ and reinforcement objective $J_{\text{RL}}(\theta^k, w^k)$.
5:     Compute gradients w.r.t. augmented parameters $\bar{\theta}^k = (\theta^k, w^k)$:
     $\nabla_{\bar{\theta}} J_{\text{SFT}}(\theta^k, w^k) = [\nabla_\theta J_{\text{SFT}}, \ \nabla_w J_{\text{SFT}}]$;
     $\nabla_{\bar{\theta}} J_{\text{RL}}(\theta^k, w^k) = [\nabla_\theta J_{\text{RL}}, \ \nabla_w J_{\text{RL}}]$.
6:     // Update lower-level variable (base parameters)
7:     $\theta^{k+1} \leftarrow \theta^k + \alpha \left[ (1 - \lambda) \nabla_\theta J_{\text{SFT}}(\theta^k, w^k) + \lambda \nabla_\theta J_{\text{RL}}(\theta^k, w^k) \right]$.
8:     // Update auxiliary $\hat{\theta}$ for upper-level (meta) gradient
9:     $\hat{\theta}^{k+1} \leftarrow \hat{\theta}^k - \alpha \nabla_\theta J_{\text{RL}}(\hat{\theta}^k, w^k)$.
10:     // Update upper-level variable (LoRA parameters)
11:     $w^{k+1} \leftarrow w^k + \beta \left[ (1 - \lambda) \nabla_w J_{\text{SFT}}(\theta^k, w^k) + \lambda \left( \nabla_w J_{\text{RL}}(\theta^k, w^k) - \nabla_w J_{\text{RL}}(\hat{\theta}^k, w^k) \right) \right]$.
12: **end for**

---

The overall algorithm of BRIDGE is presented in Algorithm 2.

### 3.3 Explanations of Update Rules

**What does the lower-level update do?** The update rule for $\theta$ in E.q. (7) is a convex combination of the SFT and RL gradients. As $\lambda$ increases from 0 to 1 during training, the algorithm gradually shifts from imitation learning to reinforcement learning.

This curriculum learning–like transition [1] is meaningful: in the early training stages, the base model lacks strong reasoning abilities and benefits more from imitating expert reasoning patterns. As training progresses, the model becomes capable of generating the correct answers by exploring the reward signal , making RL updates more valuable.

**What does the higher-level update do?** The update for $w$ in Eq. (8) aims to solve the bilevel formulation in Eq. (4). Specifically, it seeks a LoRA module $w$ such that, after training the base parameter $\theta$ on $D_{\text{RL}}$, the resulting fine-tuned model $\theta^*(w)$ also performs well on the supervised dataset $D_{\text{SFT}}$ (i.e., expert demonstrations).

The update in Eq. (8) can be interpreted as performing gradient ascent on the following objective:

$$f(\theta, w) = (1 - \lambda) \underbrace{J_{\text{SFT}}(\theta, w)}_{\uparrow \text{ likelihood on expert data}} + \lambda \underbrace{\left[ J_{\text{RL}}(\theta, w) - J_{\text{RL}}(\hat{\theta}, w) \right]}_{\uparrow \text{ reward gap between collaboration and solo}} \tag{10}$$

The first term encourages maximizing the likelihood of expert reasoning patterns in $D_{\text{SFT}}$, while the second term increases the gap between the joint optimization using both SFT and RL (parameter $\theta$), versus using only RL (auxiliary parameter $\hat{\theta}$). This contrastive gap explicitly promotes synergy between the SFT and RL objectives—ensuring that their joint optimization yields better performance than optimizing RL alone.

# 4 Experiment

## 4.1 Settings

**Datasets.**  We adopt the LIMR dataset [18] for RL training, which is derived from MATH [10]. Following their setup, we use the *Hard* subset (problems with MATH difficulty levels 3–5), which contains approximately 1.3k problems. For the SFT dataset, we follow the procedure used in DeepMath-103k [9] and distill intermediate expert reasoning steps using the R1 model.

For evaluation, we use the MATH500 subset as the primary test set and uniformly sample an additional 500 problems for validation. To assess generalization, we evaluate on a diverse set of mathematical reasoning benchmarks, including MATH500 [10], Minerva Math [16], and OlympiadBench [8], as well as competition-level datasets such as AIME 2024 and AMC 2023.

**Models.**  We conduct zero-shot RL training experiments using Qwen-2.5 models [30], selected for their demonstrated stability on mathematical reasoning tasks. The 3B model is used, with prompt formats consistent with SimpleRL.

**Reward.**  In line with SimpleRL-Zoo [31], we adopt a binary reward function based solely on answer correctness: correct final answers receive a reward of +1, while incorrect answers receive 0. We deliberately exclude format-based rewards, which have been shown to constrain exploration and reduce final performance, particularly for base models.

**Implementation Details.**  All models are trained using the `verl` framework [27] with unified hyperparameters: a prompt batch size of 64, 5 rollouts per prompt, a maximum rollout length of 3000 tokens, and a mini-batch size of 64. For evaluation, we use greedy decoding with a temperature of 0 and a maximum generation length of 5000 tokens. The learning rate is set to $5 \times 10^{-7}$, and for LoRA, both the rank and $\alpha$ are set to 16. The weighting coefficient $\lambda$ is set to 0.5. Following SimpleRL-Zoo [31], we report pass@1 accuracy for most benchmarks. For AIME 2024, due to the limited number of test cases, we additionally report average accuracy over 8 samples (avg@8). All experiments are conducted on four NVIDIA A100 GPUs (80GB).

## 4.2 Baselines

We evaluate our approach against a comprehensive set of baselines, all built on the same base architecture. These comparisons are designed to isolate the specific contributions of our proposed bilevel optimization framework.

**Base / Instruct Model.**  The original performance of base model and its instruction version, without further reasoning-specific training. This serves as a lower-bound reference for evaluating reasoning capabilities.

**Supervised Fine-Tuning (SFT).**  A model trained solely via supervised fine-tuning on curated chain-of-thought (CoT) data, without any reinforcement learning. This highlights the benefits and limitations of pure imitation-based learning.

**RL-Zero.**  Reinforcement learning applied directly to the base model without any prior fine-tuning. This baseline evaluates the effectiveness of exploration from scratch, without initialization from expert demonstrations.

**Cold-Start**  A two-stage pipeline where SFT is used to pretrain the model, followed by RL fine-tuning. The two phases are fully decoupled, with no interaction between supervised and reward-based updates.

**Naive Alternating.** A simple training procedure that alternates between SFT and RL updates in fixed intervals, without any explicit coordination or shared optimization objective between the two paradigms.

## 4.3 Main Results

| Method | MATH 500 | Minerva Math | Olympiad Bench | AIME24 (Avg@8) | AMC23 | Average |
|---|---|---|---|---|---|---|
| Base | 32.4 | 11.8 | 7.9 | 0.0 | 20.0 | 14.4 |
| Instruct | 50.8 | 14.7 | 16.7 | 8.5 | 32.5 | 24.6 |
| RL-zero | 64.4 | 26.5 | 27.0 | 3.3 | 40.0 | 32.2 |
| SFT | 53.4 | 18.8 | 21.5 | 3.3 | 42.5 | 27.9 **(-13.4%)** |
| Cold-start | **66.0** | 24.3 | 26.8 | 9.0 | 35.0 | 32.2 **(+0.0%)** |
| Naive Alter. | 65.2 | 25.3 | 27.1 | 6.7 | 42.5 | 33.4 **(+3.1%)** |
| BRIDGE | 65.4 | **28.3** | **28.4** | **10.0** | **50.0** | **36.4** **(+12.4%)** |

Table 1: Performance of our method compared to baselines methods across multiple math benchmarks. The best performance in each column is highlighted in green bold, and performance improvements (%) over RL-zero are shown in blue.

**Generalization to Benchmarks**. We evaluate the generalization ability of BRIDGE across five diverse mathematical reasoning benchmarks. As shown in Table 1, **BRIDGE** consistently outperforms baseline methods, achieving accuracy improvements of *6.8%, 12.0%, 203.0%*, and *25.0%* over RL-zero on Minerva Math, Olympiad Bench, AIME24, and AMC23, respectively. Overall, BRIDGE yields an average improvement of 12.4%, highlighting its effectiveness and robustness across tasks of varying difficulty.

Baseline methods tend to yield larger improvements on relatively easier benchmarks but generalize poorly to more complex reasoning tasks. For example, the Cold-start method underperforms RL-zero on *Minerva Math*, *Olympiad Bench*, and *AMC23*, potentially due to overfitting during the prior SFT phase. While the Naive Alternative partially mitigates this issue—maintaining performance on harder benchmarks—its gains remain limited. In contrast, BRIDGE, which explicitly encourages cooperative behavior through a reward gap mechanism, achieves consistent and substantial improvements on the more challenging benchmarks. These results underscore BRIDGE's superior generalizability in handling complex mathematical reasoning.

| Method | Average Performance | | | Average |
|---|---|---|---|---|
| | Epoch=1 | Epoch=3 | Epoch=6 | |
| RL-zero | 14.8 | 17.5 | 32.2 | 21.5 |
| SFT | 24.1 | 26.5 | 27.9 | 26.2 **(+21.8%)** |
| Cold-start | **33.4** | 28.5 | 32.2 | 31.4 **(+46.0%)** |
| Naive Alter. | 13.0 | 30.8 | 33.4 | 25.7 **(+19.5%)** |
| BRIDGE | 32.3 | **33.3** | **36.4** | **34.0** **(+69.3%)** |

Table 2: Performance progression across training epochs for different methods.

**Performance on varied fine-tuning epochs**. To evaluate *the trade-off between performance and training efficiency*, we assess the effectiveness of BRIDGE across different fine-tuning epochs. We consider the average performance across multiple epochs as a metric to reflect this trade-off. As shown in Table 2, BRIDGE achieves the best balance, with an average performance improvement of *69.3%* over RL-zere.

Among the baselines, Cold-start yields the second-best trade-off. However, its performance becomes unstable as training progresses, eventually converging to the same final result as RL-zero. In contrast, BRIDGE demonstrates consistent improvement throughout training. Overall, nearly all hybrid baselines outperform RL-zero in terms of early-stage efficiency, highlighting the advantage of integrating supervised fine-tuning and reinforcement learning paradigms.

# 5 Related Work

**Reinforcement Learning for Large Reasoning Models.** Recent progress has highlighted the critical role of reinforcement learning in enhancing the reasoning capabilities of large language models [21, 7]. DeepSeek-R1 introduced a simple yet effective rule-based reward model and demonstrated further gains through multiple rounds of supervised distillation and RL training. LIMR [18] showed that complex reasoning behaviors can emerge from as few as one thousand curated examples from the MATH dataset [11].

In parallel, substantial advances have been made in training recipes for large reasoning models. Chu et al. [4] compare SFT and RL for reasoning tasks and find that RL generalizes significantly better, whereas SFT is prone to overfitting. SimpleRL [31] observes that fine-tuning on short-CoT datasets can harm reasoning ability, while He et al. [9] find that fine-tuning on long-CoT distilled data can improve the reasoning performance of smaller models—especially when used as a warm-up stage before RL training. In practice, two-stage pipelines that combine SFT and RL are commonly used to balance stability and performance. However, existing approaches often rely solely on supervised fine-tuning, which tends to generalize poorly, or on pure RL, which suffers from sample inefficiency and unstable optimization. In this work, we propose the first unified training framework that enables explicit interaction between SFT and RL via a bilevel optimization formulation. This approach offers a new perspective on integrating imitation and exploration for reasoning-centric large language models.

**Bilevel Optimization in LLMs.** Bilevel optimization (BLO) is a classical framework for modeling nested learning problems, where an upper-level objective depends on the solution to a lower-level optimization task. Two major classes of methods have been developed to solve BLO problems. Implicit gradient methods [12, 14, 23, 29] compute gradients through the lower-level problem using second-order derivatives. While theoretically robust, these methods are often computationally expensive and memory-prohibitive when applied to large-scale models such as LLMs. In contrast, penalty-based relaxation methods [24, 15, 25, 20] approximate the BLO formulation using only first-order gradients, making them substantially more scalable and thus better suited for LLM applications. Recent work has explored the use of bilevel optimization in LLMs for tasks such as data selection [19, 26], inverse reinforcement learning [17], and meta-learning [3, 28]. To the best of our knowledge, our work is the first to apply bilevel optimization to reasoning-oriented LLM training, providing a principled approach to integrating supervised and reinforcement learning in a unified framework.

# 6 Conclusion

This work investigates how to effectively integrate supervised fine-tuning and reinforcement learning to improve the reasoning capabilities of large language models. We begin by analyzing three widely used training paradigms and identify a key limitation of existing multi-stage pipelines: the lack of interaction between SFT and RL. To address this, we propose a simple alternating baseline and further introduce *BRIDGE*, a bilevel optimization framework that models SFT as the upper-level objective and RL as the lower-level objective. By employing a penalty-based relaxation, BRIDGE explicitly encourages joint training to outperform standalone RL, fostering tighter synergy between the two learning paradigms. Empirical results on six mathematical reasoning benchmarks demonstrate that our method consistently outperforms strong baselines in both accuracy and training efficiency. These findings underscore the potential of bilevel optimization as a unifying framework for combining supervised and reward-driven learning in complex reasoning tasks.

**Limitations.** While BRIDGE demonstrates promising results, it introduces additional computational overhead due to its nested bilevel optimization structure. Future work includes extending the framework to larger-scale models and more diverse domains such as program synthesis, theorem proving, and scientific reasoning, as well as exploring more efficient optimization strategies to mitigate the computational cost.

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
