# OpenReview forum: "Beyond Two-Stage Training: Integrating SFT and RL for Improved Reasoning in LLMs"
_NeurIPS.cc/2025/Conference — Submitted to NeurIPS 2025_

### Official Review · Reviewer_qL6k · 2025-06-08

**Clarity:** 3
**Significance:** 2
**Originality:** 3
**Rating:** 4
**Confidence:** 4

**Summary:**

This paper proposes combining two-stage fine-tuning—SFT followed by RL—using a bi-level optimization framework. The lower-level optimization enables RL-based exploration constrained by SFT supervision, while the upper-level optimization ensures that the joint training achieves higher rewards than RL alone.

**Questions:**

If the authors could provide the necessary additional experiments (covering multiple model families, model sizes, and in-distribution performance) along with more in-depth analysis (such as evaluating the method with different RL techniques, particularly DPO variants), I would be happy to raise my score.

**Ethical Concerns:**

["NO or VERY MINOR ethics concerns only"]

**Final Justification:**

The authors initially demonstrated the effectiveness of their method using only a single backbone model and one in-distribution task, which was insufficient to fully validate the proposed approach. Though the idea of jointly optimizing RL and SFT is interesting. In the revision, additional experiments were provided across multiple in-distribution datasets, using different backbone models and integrating various RL algorithms. Overall, the method shows promising performance on models in the 7B–8B scale. However, its effectiveness on larger models remains uncertain. Therefore, the final score is 4.

**Limitations:**

yes

**Quality:**

2

**Strengths And Weaknesses:**

## Pros:

1. I am not an expert in bi-level optimization, but I find the idea to be interesting. The paper effectively bridges SFT and RL fine-tuning, two widely used but typically separate approaches in current LLM training pipelines (often used with a cold-start).
2. The paper is well written and easy to follow.
3. The motivation is sound, and the topic is timely and relevant.

## Cons:

The main weakness of this paper is the lack of extensive experiments and in-depth analysis.

1. The framework is only evaluated on a single LLM (Qwen3.5-3B). To convincingly demonstrate generalization, the authors should conduct experiments across multiple model families (e.g., LLaMA) and model sizes.
2. While the paper shows out-of-distribution generalization at inference time on other benchmarks, the model is trained on only one dataset. To further support claims of generalization, the authors should train on multiple datasets and report in-distribution performance.
3. The authors claim the method outperforms vanilla RL approaches. However, RL-based fine-tuning encompasses several families of methods, with PPO, DPO, and their variants being among the most commonly used. The paper primarily discusses GRPO/PPO. It remains important to evaluate whether this bi-level optimization technique can be seamlessly integrated with DPO-style methods as well.

## Typo:

Line 247: "RL-zere" should be corrected to "RL-zero."

---

> ### Author Rebuttal · Authors · 2025-07-31
>
> Thank you for your thoughtful and constructive feedback on our work. We are encouraged that you found our motivation sound, idea interesting, and paper well-written. We have conducted extensive additional experiments and analyses to address your primary concerns regarding the experiments.
>
> ## Summary of Additional Experiments
>
> In response to your suggestions, we have:
>
> 1.  **Extended experiments to additional LLMs**: We tested on two additional models:  We evaluated BRIDGE on _Qwen3-8B-Base_ and _Llama-3.2-3B-Instruct_, and BRIDGE achieved improvements of 13% and 31% over RL-zero and Cold-start respectively on LLaMA-3.2-3B, and 16% and 10% improvements on Qwen3-8B-Base.
>
> 2.  **Added in-distribution experiments**: We trained and evaluated on three datasets with in-distribution testing. BRIDGE demonstrated consistent improvements of 6% over RL-zero and 3% over Cold-start.
>
> 3.  **Integrated DPO-style training**: We verified compatibility with DPO algorithms, where BRIDGE achieved a 39% improvement over RL-zero, confirming our bi-level framework is algorithm-agnostic.
>
> Below, we address each point in detail.
>
> ***
>
>
> ## Response to W1: Experiments on Additional LLMs
>
> We appreciate your suggestion. To assess generalization across model families and sizes, we conducted additional experiments on **Qwen3-8B-Base** and **LLaMA-3.2-3B-Instruct**, beyond the original **Qwen2.5-3B**.
>
> Tables 1 and 2 present these results. On Qwen3-8B-Base, BRIDGE achieves an average accuracy score of 49.9, representing a 16% improvement over RL-zero and a 10% improvement over Cold-start. On LLaMA-3.2-3B-Instruct, BRIDGE outperforms RL-zero by 13% and Cold-start by 31%, demonstrating strong generalization across model families.
>
> Table 1. Experiments on Qwen3-8B-Base
> |  | math500 | minerva math | Olympiad Bench | AIME24  | AMC23 | Avg. |
> | :--- | :--- | :--- | :--- | :--- | :--- | :--- |
> | Base | 55.4 | 24.3 | 22.5 | 3.3 | 27.5 | 26.6 |
> | RL-zero | 76.2 | 36 | 42.4 | 10 | 50 | 42.9 |
> | Cold-start | **80.4** | 38.2 | 39.6 | 16.6 | 52.5 | 45.5 |
> | Bridge | 79.0 | **39.7** | **44.0** | **16.7** | **70.0** | **49.9** |
>
> Table 2. Experiments on Llama-3.2-3B-Instruct
> |  | math500 | minerva math | Olympiad Bench | AlME24 | AMC23 | Avg. |
> | :--- | :--- | :--- | :--- | :--- | :--- | :--- |
> | Instruct | 38 | 14.3 | 13 | 13.3 | 25 | 20.7 |
> | RL-zero | 48.6 | **15.1** | 17.8 | **10.0** | 17.5 | 21.8 |
> | Cold-start | 45 | 11.8 | 12 | 3.3 | 22.5 | 18.9 |
> | Bridge | **51.8** | **15.1** | **19.3** | **10.0** | **27.5** | **24.7** |
>
> ## Response to W2: In-Distribution Evaluation
> Thank you for the valuable suggestion. To further support our generalization claims, we conducted additional training on three datasets: **GSM8K**, **MATH (Level 3–5)**, and **DeepMath**, each with 3k training examples. We then evaluated performance on their respective test sets.
>
> Table 3. In-Distribution Performance Across Three Datasets
> |  | GSM8k | Math level3-5 | DeepMath | Avg |
> | :--- | :--- | :--- | :--- | :--- |
> | RL-zero | 92.8 | 76.7 | 86.2 | 85.2 |
> | Cold-start | 93.1 | 80.2 | 88.5 | 87.3 |
> | BRIDGE | 94.3 | 84.6 | 91.8 | 90.2 |
>
> As shown in Table 3, BRIDGE consistently outperforms the baselines on all datasets, demonstrating robust in-distribution generalization in addition to out-of-distribution performance.
>
> ## Response to W3: Compatibility with DPO-Style Methods
>
> Thank you for raising this important point. Our bi-level framework is **algorithm-agnostic** and can be integrated with a variety of RL methods.
>
> To demonstrate this flexibility, we conducted experiments using **DPO** as the underlying RL algorithm. As shown in Table 4, BRIDGE achieves a 39% relative improvement over DPO-zero,  confirming compatibility beyond PPO/GRPO paradigms.
>
> Table 4. Experiments on DPO
> |  | math500 | minerva math | Olympiad <br> Bench | AIME24 | AMC23 | Avg. |
> | :---: | :---: | :---: | :---: | :---: | :---: | :---: |
> | DPO-zero | 36.0 | 12.5 | 10.0 | 0.0 | 25.5 | 16.8 |
> | DPO-Bridge | 48.0 | 14.7 | 15.8 | 3.3 | 35.0 | 23.4 |
>
>
> ### **Minor Correction**
>
> Thank you for pointing out the typo. We will correct "RL-zere" to "RL-zero" in Line 247.
>
> ***
> We sincerely appreciate your thoughtful and constructive feedback, which has substantially strengthened our evaluation. We hope these additional results adequately address your concerns and welcome any further questions or suggestions.

---

> ### Author Response · Authors · 2025-08-03
> **Response to Your Feedback**
>
> Dear Reviewer qL6k,
>
> Thank you again for your valuable and constructive feedback. We have conducted the additional experiments you suggested, including:
>
> 1. **Multiple model families and sizes**: We extended our experiments to Qwen3-8B-Base and LLaMA-3.2-3B-Instruct, showing consistent improvements (13–31%) over baselines.
>
> 2. **In-distribution evaluation**: We trained and evaluated on GSM8K, MATH, and DeepMath datasets, achieving 6% improvement over RL-zero.
>
> 3. **DPO compatibility**: We successfully integrated our framework with DPO, yielding a 39% improvement over DPO-zero.
>
> We sincerely hope these results address your concerns regarding the evaluation scope. With the discussion period ending on August 6, we would greatly appreciate it if you could take a moment to review our updates and share your thoughts.
>
> Thank you again for your time and consideration.
>
> Best regards,
> The authors

---

> ### Comment · Reviewer_qL6k · 2025-08-03
>
> I appreciate the author's efforts on providing more experiments in supporting the effectiveness of the algorithm, also recommend the authors to incorporate these results in their future revisions. I will improve my score accordingly.

---

> > ### Author Response · Authors · 2025-08-04
> > **Thank You for Your Feedback and Recognition**
> >
> > Thank you for your feedback on our additional experiments and for your positive assessment. Your constructive suggestions throughout the review process have really improved our work. We appreciate your time and effort.

---

### Official Review · Reviewer_XEdK · 2025-06-20

**Clarity:** 3
**Significance:** 3
**Originality:** 3
**Rating:** 5
**Confidence:** 3

**Summary:**

A novel training paradigm that combines SFT and RL.

**Questions:**

(1) The paper introduce LoRA during training, does other baseline method uses LoRA training as well?

(2) How much additional training time cost does the proposed training introduce compared to SFT or RL training?

(3) Follow (2), how many additional computational cost is required to implement the proposed bilevel optimization, given the complexity of optimizing the model following two policy, I wonder if the training efficiency and performance tradeoff is worthwhile.

**Ethical Concerns:**

["NO or VERY MINOR ethics concerns only"]

**Final Justification:**

The author's response justify that the proposed method, compared with SFT, does not incur additional computational overhead, which address my concerns. I will raise my score to 5: accept.

**Limitations:**

The proposed training strategy is interesting and effective, and seems that there is no too much limitations beside the training complexity issue.

Given the complexity of the bilevel optimization, I believe the paper's main limitation lies in how to efficiently train the LLM. Very likely the proposed training will introduce extra complexity during training, and I hope the author can provide more analysis and discussion on this. See questions.

**Quality:**

3

**Strengths And Weaknesses:**

- The proposed training method is interesting, with a strong motivation of taking the advantages from both SFT and RL training.
- Paper is well written with clear method explained.
- Performance improvements over pure SFT, RL and other training methods is notable on several public benchmarks.

---

> ### Author Rebuttal · Authors · 2025-07-31
>
> Thank you for your positive and insightful feedback on our work. We are encouraged that you found the method interesting, the motivation strong, and the performance improvements notable across benchmarks. We appreciate your recognition of the clear method explanation and effective combination of SFT and RL advantages.
>
> ## Summary of Response
>
> In response to your questions, we have:
>
> 1.  **Clarified LoRA usage**: We confirmed that baseline methods do not use LoRA and demonstrated through ablation that LoRA's impact on baseline performance is minimal (< 0.2% difference).
>
> 2. **Provided comprehensive cost-benefit analysis**: BRIDGE demonstrates superior training efficiency, achieving **_44% faster training_** than Cold-start on Qwen2.5-3B with **_13% better performance_**, and **_14% faster training_** than Cold-start on Qwen3-8B-Base with **_10% better performance_**. While BRIDGE introduces moderate memory overhead, these significant improvements in both efficiency and accuracy justify the trade-off.
>
>
>
> ## Response to Q1: Use of LoRA in Baselines
> Thank you for your question. All baseline methods in our paper are trained using the base model only, without LoRA. To assess the influence of LoRA on performance, we also trained the baselines with LoRA, which account for only 0.5% of the total parameters. As shown in Table 1, the performance impact is minimal.
>
> Table 1. Effect of LoRA on RL-zero
> | Method | Score |
> | :--- | :--- |
> | RL-zero | 32.2 |
> | RL-zero + LoRA | 32.0 |
>
>
> ## Reponse to Q2: Cost-Benefit Analysis
>
> Thank you for this insightful suggestion. We agree that averaging performance across epochs is not a standard metric for evaluating the performance-efficiency trade-off. In response, we conducted a comprehensive cost-performance analysis reporting **wall-clock training time**, **average GPU memory usage per device**, and **final convergence performance** across two model scales: **Qwen2.5-3B** (trained on 4×80GB A100 GPUs) and **Qwen3-8B-Base** (trained on 8×192GB MI300 GPUs).
>
> Table 2. Cost-performance analysis of Qwen 2.5-3B
> |  | RL-zero | Cold-start | BRIDGE |
> | :--- | :--- | :--- | :--- |
> | Clock Time (Hour) | 6.1 | 12.3 | 6.9 |
> | GPU Memory (GB) | 52.2 | 45.9 | 59.3 |
> | Performance (Acc.) | 32.2 | 32.2 | 36.4 |
>
> Table 3. Cost-performance analysis of Qwen 3-8B-Base
> |                      | RL-zero | Cold-start | BRIDGE |
> |----------------------|---------|------------|--------|
> | Clock Time (Hour)    | 38.5    | 39.1       | 33.5   |
> | GPU Memory (GB)      | 50.7    | 60.8       | 67.4   |
> | Performance (Acc.)   | 42.9    | 45.5       | 49.9   |
>
> ### Key Findings:
>
> 1.  **Cold-start suffers from significant inefficiency** due to _**data distribution mismatch**_ between SFT and RL stages. During SFT, the model learns from teacher-distilled data (R1 model), producing long reasoning traces. This leads to excessive response lengths during early RL training, increasing computational costs. We observe a "dip-then-rise" pattern in response length, suggesting that _the model first somewhat forgets the expert behavior acquired during SFT and then slowly explores new behaviors_. This contributes to the inefficiency and suboptimal performance of cold-start training.
>
> 2.  **BRIDGE achieves superior training efficiency** by dynamically integrating both SFT and RL data distributions within a single-stage process. This design mitigates the data distribution and response-length mismatch problem in the two-stage pipeline, resulting in:
>     -   **44% faster training** than Cold-start on Qwen2.5-3B with **13% better performance**
>     -   **14% faster training** than Cold-start on Qwen3-8B with **10% better performance**
>     -   Even outperforming RL-zero by **13% in speed** on larger models
> 3.  **The computational trade-off is favorable**: While BRIDGE introduces additional memory overhead, the substantial gains in both training efficiency and final performance justify this cost.
>
>
> ## Reponse to Q3:  Computational Cost Comparison
> Please refer to our response to W1 for detailed computational cost analysis.
>
>
> ***
> We sincerely appreciate your thoughtful comments and suggestions, which have significantly enhanced our work. We hope our responses and new results effectively address your concerns, and we welcome any further feedback you may have. Thank you again for your review.

---

> ### Author Response · Authors · 2025-08-03
> **Thank You for Your Review - Efficiency and LoRA Analysis Added**
>
> Dear Reviewer XEdK,
>
> Thank you for your positive and constructive review. We're glad you found the method interesting and well-motivated, and we appreciate your thoughtful questions regarding training efficiency and implementation complexity.
>
> Following your suggestions, we conducted a detailed cost-benefit analysis across two model scales. Despite the use of bilevel optimization, BRIDGE achieves 44% faster training than Cold-start on Qwen2.5-3B and 14% faster on Qwen3-8B, while also delivering 13% and 10% better final performance, respectively. We believe this demonstrates a favorable trade-off between efficiency and accuracy in practice.
>
> We also clarified that baseline methods do not use LoRA. Ablation experiments show that adding LoRA leads to negligible performance change (<0.2%), confirming that LoRA is not a confounding factor in our comparisons.
>
> With the discussion phase ending on August 6, we just wanted to ensure you had a chance to see these new results. If there's anything else we can clarify, we'd be very happy to discuss further.
>
> Thank you again for your careful review--it has helped us strengthen the paper.
>
> Best regards,
>
> The authors

---

### Official Review · Reviewer_1amM · 2025-06-26

**Clarity:** 3
**Significance:** 2
**Originality:** 3
**Rating:** 4
**Confidence:** 3

**Summary:**

This paper proposes BRIDGE, a bilevel optimization framework that unifies commonly used decoupled pipeline for training reasoning models: iteratively using SFT and RL. In BRIDGE, SFT is formulated as the upper-level optimization and RL as the lower-level. The method is evaluated on six mathematical reasoning benchmarks, showing performance gains over iterative SFT-RL baselines.

**Questions:**

1. What is the fundamental challenge with the decoupled (alternating) training of SFT and RL that motivates the bilevel formulation?
2. In the BRIDGE framework, why is SFT treated as the upper-level objective and RL as the lower-level? Would reversing the roles lead to worse results, or is it simply an arbitrary design choice?
3. The learning outcome is undoubtedly important, but the BRIDGE method has profoundly changed the learning dynamics. Presenting the learning dynamics might bring more insights to this community. Could you provide a comparison of the BRIDGE and Naive Alternating training curves, including rewards and KL penalties?

**Ethical Concerns:**

["NO or VERY MINOR ethics concerns only"]

**Final Justification:**

My concerns have been addressed.

**Limitations:**

Yes.

**Paper Formatting Concerns:**

No formatting concern.

**Quality:**

2

**Strengths And Weaknesses:**

### Strength
The paper presents an original and intuitive formulation to integrate SFT and RL. The paper is well-writen and easy to follow. The theoretical formulation is clear. Empirical results across multiple benchmarks show promising improvements on learning outcomes.

### Weakness
In terms of experimental evaluation, since all baselines were implemented by the authors, the reported implementation details are somewhat insufficient. Only settings such as PPO parameters, LoRA configuration are briefly described. The specific implementations of Cold-Start, Naive Alternating, and BRIDGE lacked sufficient details (such as the SFT-RL switching intervals in Naive Alternating, the penalty strength $\gamma$ in BRIDGE, etc.).

---

> ### Author Rebuttal · Authors · 2025-07-31
>
> Thank you for your thoughtful feedback on our work. We appreciate your recognition of our method as original and intuitive, with clear theoretical formulation and promising empirical results. We are pleased that you found our paper well-written and easy to follow.
>
> ## Summary of Responses
>
> We have addressed your concerns by:
>
> 1.  **Providing comprehensive implementation details** for all baselines and BRIDGE, including previously unspecified hyperparameters
> 2.  **Clarifying the fundamental challenge** that motivates our bilevel formulation: the data distribution mismatch in two-stage training
> 3.  **Explaining the role assignment** in BRIDGE based on the natural flow from SFT initialization to RL optimization
> 4.  **Analyzing learning dynamics** through comparative reward and KL penalty trajectories
>
> Below, we address each point in detail.
>
> ***
>
> ## Response to W1: Implementation Details
> We appreciate the opportunity to clarify the implementation details. Here are the complete configurations:
>
>
> **Cold-Start (Two-Stage Pipeline)**
>
> -   SFT dataset: Distilled from R1 model
> -   SFT stage: Batch size = 32, learning rate = 1e-5, epochs = 2 (best validation checkpoint selected after 1 epoch)
> -   RL stage: Batch size = 64, mini-batch size = 64, learning rate = 5e-7, epochs = 6, rollouts = 5, maximum sequence length = 5,000
>
> **Naive Alternating (Algorithm 1)**
>
> -   SFT step: Batch size = 64, learning rate = 1e-5
> -   RL step: Batch size = 64, mini-batch size = 64, learning rate = 5e-7
> -   Switching interval: 1 iteration (alternating every iteration)
>
> **BRIDGE (Algorithm 2)**
>
> -   SFT: Batch size = 64, learning rate = 1e-5
> -   RL: Batch size = 64, mini-batch size = 64, learning rate = 5e-7
> -   Penalty coefficient: λ = 0.5
> -   LoRA configuration: Rank = 16, α = 16
> - All other hyperparameters are identical to those used in the Naive alter
>
> Additional training details are provided in Section 4.1. We will incorporate these specifications into our revised manuscript and release our code for reproducibility.
>
> ***
>
> ## Response to Q1: Challenge of the Two-Stage Pipeline
>
> Thank you for the insightful question. The primary limitation of the two-stage paradigm is the **data distribution mismatch between SFT and RL stages**.
>
> In the first stage, the SFT data $D_{sft}$ is distilled from a stronger teacher model (R1), producing long and well-structured reasoning traces. In contrast, the RL data $D_{rl}$in the second stage is sampled from the student model itself, which initially generates shorter and less complete responses.
>
> This mismatch creates two critical issues:
>
> 1. **Training inefficiency**: Early in RL training, the model tends to generate overly long responses, increasing computational cost. We observe that the response length initially drops, then gradually recovers as training progresses.
>
> 2. **Distribution shift**: This “dip-then-rise” pattern suggests that _the model first somewhat forgets the expert behavior acquired during SFT and then slowly explores new behaviors_. This mismatch contributes to the inefficiency and suboptimal performance of cold-start training.
>
> Table 1. Training Time and Performance Comparison
> |  | Cold-start | BRIDGE |
> | :--- | :--- | :--- |
> | Clock Time (Hour) | 12.3 | 6.9 |
> | Performance (Acc.) | 32.2 | 36.4 |
>
> As shown in Table 1, these effects result in nearly 2× longer training time and 13% lower performance compared to BRIDGE.  We believe this stems from the lack of interaction between the SFT and RL stages, which prevents smooth adaptation. This observation motivates our design of a more integrated training framework.
>
>
> ## Response to Q2: Role Assignment in BRIDGE
>
> Our role assignment follows the natural training flow where SFT provides initialization for RL optimization. In the standard pipeline, SFT creates a strong starting point for subsequent RL training.
>
> BRIDGE extends this principle through meta-learning: at each iteration, upper-level SFT provides an improved initialization for lower-level RL exploration. This design preserves the established SFT→RL flow while enabling continuous interaction.
>
> Reversing the roles (RL as upper-level, SFT as lower-level) would contradict this fundamental principle, as it would imply RL provides initialization for SFT—contrary to the general principle.
>
>
> ## Response to Q3: Learning Dynamics Analysis
>
>
> Following your suggestion, we analyzed the reward trajectories and KL penalties for BRIDGE versus Naive Alternating. Tables 2 and 3 present these dynamics.
>
> Table 2. Mean Reward Dynamics
> |  | 0 | 20 | 40 | 60 | 80 | 100 | 120 |
> | :--- | :--- | :--- | :--- | :--- | :--- | :--- | :--- |
> | Naїve Alter. | 0.05 | 0.03 | 0.08 | 0.20 | 0.27 | 0.37 | 0.36 |
> | BRIDGE | 0.05 | 0.07 | 0.31 | 0.36 | 0.45 | 0.49 | 0.51 |
>
> Table 3. KL Penalty Dynamics
> |  | 0 | 20 | 40 | 60 | 80 | 100 | 120 |
> | :--- | :--- | :--- | :--- | :--- | :--- | :--- | :--- |
> | Naïve Alter. | 0 | 0.002 | 0.006 | 0.02 | 0.03 | 0.04 | 0.04 |
> | BRIDGE | 0 | 0.1 | 0.04 | 0.06 | 0.06 | 0.07 | 0.05 |
>
> Our analysis reveals that BRIDGE demonstrates more efficient exploration in the early training stages, with faster reward growth accompanied by larger KL penalties compared to the Naive Alternating approach. As training progresses toward completion, BRIDGE achieves higher average rewards while the KL penalties of both methods converge to similar levels.
>
> These findings suggest that BRIDGE's training dynamics enable more effective policy optimization through better balancing of exploration and constraint satisfaction. We will incorporate these comparative learning curves and discussion in our revised manuscript to provide the community with deeper insights into the underlying mechanisms.
>
>
> ***
> We sincerely appreciate your constructive feedback, which has substantially strengthened our work. We hope these additional details and analyses adequately address your concerns and welcome any further questions.

---

> ### Author Response · Authors · 2025-08-03
> **Response to Your Review Comments**
>
> Dear Reviewer 1amM,
>
> Thank you again for your insightful review and thoughtful questions. We have carefully responded to your comments in our rebuttal, including:
>
> 1. **Implementation details**: We provided comprehensive configurations for all baselines (Cold-Start, Naive Alternating, and BRIDGE), including previously unspecified hyperparameters.
>
> 2. **Learning dynamics**: We presented an analysis of reward trajectories and KL penalties between BRIDGE and Naive Alternating, highlighting how BRIDGE enables more efficient early-stage exploration and achieves higher final rewards.
>
> 3. **Bilevel motivation**: We clarified the fundamental challenge of data distribution mismatch between SFT and RL stages, which leads to longer training time and suboptimal performance in traditional pipelines.
>
> We hope these responses can address your concerns. With the discussion phase ending on August 6, we would greatly appreciate any thoughts or feedback you might have on our responses. Please feel free to let us know if there are any remaining questions we can help clarify.
>
> Thank you again for your time and engagement.
>
> Best regards,
> The authors

---

> ### Author Response · Authors · 2025-08-05
> **Looking forward to your reply**
>
> Dear Reviewer 1amM,
>
> Thank you for your efforts in reviewing our work and for your insightful questions. Following your suggestions, we have elaborated on our implementation details and discussed the challenges of the two-stage baseline, the motivation behind our design, and the learning dynamics. We truly hope that our response helps clarify these issues.
>
> As the discussion deadline is approaching, please let us know if there is anything else you need further information on or any additional concerns you might have.
>
> Best regards,
>
> Authors

---

> > ### Comment · Reviewer_1amM · 2025-08-06
> >
> > Sorry for the delayed response, I spent a lot of time reading other reviewers' feedbacks. I appreciate the authors’ effort in providing additional experimental results and detailed explanations. My concern regarding Q2 has been adequately addressed. However, I still find the motivation behind the proposed method somewhat unconvincing.
> >
> > From my understanding, let's assume the optimal policy for SFT is $\pi_{\theta_1^\*}$, while that for RL is $\pi_{\theta_2^\*}$. Naive Alternating switches gradient directions between the two objectives, whereas BRIDGE‘s gradient (informally speaking)  takes both ojectives into consideration in each update. While this idea is reasonable, I remain unsure why such an approach would necessarily yield better performance, especially compared to Cold-Start, which directly optimizes toward $\pi_{\theta_2^\*}$ and aligns more directly with the evaluation criterion (accuracy).
> >
> > My main concern is: what specific advantages does BRIDGE bring by simultaneously considering $\pi_{\theta_1^\*}$ and $\pi_{\theta_2^\*}$ at each step, as opposed to Cold-Start’s more focused optimization toward $\pi_{\theta_2^\*}$?
> >
> > I would appreciate further clarification, and I remain open to being convinced by additional insights or evidence.

---

> ### Author Response · Authors · 2025-08-06
> **Clarification on the Motivation of BRIDGE**
>
> Thank you for your thoughtful feedback and for engaging deeply with our work. We appreciate your openness to further discussion.
>
> We'd like to clarify our approach from two complementary perspectives and extended empirical evidence:
>
>
> ## 1. From RL Perspective: Addressing Exploration Challenges
>
> **Q: Why introduce SFT?**
>
> RL optimization faces fundamental challenges: (1) due to the unknown environment and non-convex optimization, practical RL cannot reach the optimal π_θ2* and often becomes trapped in local optima; (2) it relies on inefficient trial-and-error to discover solutions that yield positive rewards.
>
> In hard reasoning problems, agents exploring independently often get stuck, unable to find reward-yielding answers. Even with SFT initialization, RL frequently reaches performance plateaus—particularly on difficult questions where agents fail to produce any positive solutions.
>
> BRIDGE addresses this by providing *adaptive* SFT guidance, **helping agents discover solution paths they cannot find alone**. This design **avoids inefficient exploration and provides external guidance to escape local optima**, with empirical results showing significant performance enhancement.
>
> ## 2. From Meta-Learning Perspective
>
> **Q: What is the meta-learning objective?**
>
> BRIDGE does not simply *optimize toward π_θ1 and π_θ2 simultaneously*. Instead, BRIDGE follows a meta-learning structure: *we optimize SFT specifically to help RL converge better to π_θ2*.
>
> Since SFT is not always beneficial for RL—naively alternating SFT and RL updates **does not guarantee that every SFT update will improve the RL objective**. To address this, BRIDGE alternates between: (1) updating LoRA adapter weights to **explicitly maximize the reward gap** between joint SFT + RL training and RL alone, and (2) updating base model parameters following a blend of SFT and RL gradients.
>
> **Q: Why is single-stage meta-learning superior to two-stage Cold-Start?**
>
> The two-stage cold-Start follows a naive continual learning paradigm with inherent *catastrophic forgetting*—the model loses valuable SFT knowledge during RL training. In contrast, BRIDGE's meta-learning paradigm:
>
> (1) Avoids forgetting: Single-stage training prevents the wasteful forgetting-relearning cycle
>
> (2) Provides better initialization: ***Each SFT update is specifically designed to provide initialization points that help RL converge toward π_θ2****
>
> The key insight: **SFT's role in BRIDGE is instrumental—it exists to facilitate RL's optimization toward π_θ2***.
>
> ## 3. Empirical Evidence
>
> To validate BRIDGE's advantages, we conduct additional experiments on **Qwen3-8B** and **LLaMA-3.2-3B-Instruct**, in addition to the previously evaluated Qwen2.5-3B.
>
> | Model | Cold-Start | BRIDGE | Improvement |
> |-------|------------|--------|-------------|
> | Qwen2.5-3B | 32.2 | 36.4 | +13% |
> | Qwen3-8B-Base | 45.5 | 49.9 | +10%|
> | Llama-3.2-3B-Instruct | 18.9 | 24.74 | +31% |
>
> These consistent improvements across diverse LLMs demonstrate that BRIDGE's meta-learning approach enables *better exploration* and *guaranteed helpfulness of SFT updates to RL objectives*, leading to superior performance compared to Cold-Start's direct but less efficient optimization.
>
> ***
> We hope these clarifications address your concerns and welcome any further questions.

---

> > ### Author Response · Authors · 2025-08-07
> > **Looking forward to your reply**
> >
> > Dear Reviewer 1amM,
> >
> > Thank you again for your thoughtful comments. We're glad to hear that some of your earlier concerns have been addressed.
> >
> > As the discussion phase will end in less than two days, we would greatly appreciate it if you could let us know whether our clarification on *BRIDGE's motivation* addressed your question, or if there are any remaining points we can further discuss. We value your feedback and are happy to continue the discussion.
> >
> > Best regards,
> >
> > Authors

---

> > ### Author Response · Authors · 2025-08-08
> > **Looking forward to your reply**
> >
> > Dear Reviewer 1amM,
> >
> > With the discussion ending tomorrow, we would appreciate knowing whether our response about BRIDGE's motivation addressed your question. Thank you for your time.
> >
> > Best regards,
> >
> > Authors

---

> > > ### Comment · Reviewer_1amM · 2025-08-09
> > >
> > > My concerns have been addressed. I'm happy to increase my score accordingly.

---

> > > > ### Author Response · Authors · 2025-08-09
> > > > **Thank You – Concerns Addressed**
> > > >
> > > > Thank you very much for your positive feedback and for confirming that our rebuttal effectively addressed your concerns.
> > > >
> > > > We sincerely appreciate your thoughtful review and continued engagement with our work. We will incorporate the suggestions in the revision. Thank you again for your support.

---

### Official Review · Reviewer_73wX · 2025-07-01

**Clarity:** 3
**Significance:** 3
**Originality:** 3
**Rating:** 4
**Confidence:** 4

**Summary:**

This paper proposes BRIDGE, a novel bilevel optimization framework to integrate SFT and RL for improving reasoning capabilities in LLMs.
The method aims to overcome the limitations of traditional decoupled two-stage training by facilitating better cooperation between SFT and RL objectives.
BRIDGE formulates SFT as an upper-level objective and RL as a lower-level objective, leveraging a penalty-based relaxation to explicitly encourage joint training to yield higher rewards than RL alone.
Empirical evaluations on five mathematical reasoning benchmarks demonstrate that BRIDGE consistently outperforms baselines in both accuracy and training efficiency.

**Questions:**

* Figure 1, which aims to compare training methods, notably lacks a dedicated curve for "SFT" alone. This makes it challenging to directly observe and confirm the claim in line 105 that "SFT exhibits rapid initial learning". While "Cold start" begins with an SFT phase, a separate SFT curve would provide clearer evidence for the initial learning dynamics.

**Ethical Concerns:**

["NO or VERY MINOR ethics concerns only"]

**Final Justification:**

Most of my concerns, especially the evaluation of performance and efficiency, is resolved. I choose to maintatin my original score.

**Limitations:**

yes

**Quality:**

3

**Strengths And Weaknesses:**

Strengths:
* Integrating SFT and RL is a noval approach to address the decoupled nature of existing multi-stage pipelines. This provides a more sophisticated interaction between imitation learning and reward-driven exploration.
* The experimental results consistently show that BRIDGE outperforms baselines across various mathematical reasoning benchmarks.
* The paper provides a clear comparative analysis of existing fine-tuning strategies, highlighting the complementary advantages of SFT and RL and motivating the need for their deeper integration.

Weaknesses:
* In Table 2, the "Average Performance across epochs" is used to evaluate the trade-off between performance and training efficiency. Averaging performance across different epochs is generally not a standard or robust metric for this trade-off. A more appropriate evaluation would involve plotting performance curves over time (epochs/steps) for all methods and comparing areas under the curve, or explicitly reporting "time-to-reach-a-target-performance" metrics.
* A more detailed discussion or empirical comparison of the computational cost of BRIDGE against baselines are needed.
* Typo:
  * The usage of λ and γ in Section 3.2 appears inconsistent.
  * line 247, zere->zero.

---

> ### Author Rebuttal · Authors · 2025-07-31
>
> Thank you for your positive and constructive feedback on our work. We are encouraged that you found the method novel, the motivation sound, and the performance consistently strong across baselines. We appreciate your recognition of BRIDGE's sophisticated integration of imitation learning and reward-driven exploration. Your main concern regarding training efficiency is well-taken, and we have conducted comprehensive analysis to address it, as detailed below.
>
>
> ## Summary of Response
>
> - **Provided comprehensive cost-benefit analysis**: BRIDGE demonstrates superior training efficiency, achieving **44% faster training** than Cold-start on Qwen2.5-3B with **13% better performance**, and **14% faster training** than Cold-start on Qwen3-8B with **10% better performance**. While BRIDGE introduces moderate memory overhead, these significant improvements in both efficiency and accuracy justify the trade-off.
>
> ***
>
> ## Reponse to W1: Cost-Benefit Analysis
>
> Thank you for this insightful suggestion. We agree that averaging performance across epochs is not a standard metric for evaluating the performance-efficiency trade-off. In response, we conducted a comprehensive cost-performance analysis reporting **wall-clock training time**, **average GPU memory usage per device**, and **average final convergence performance** across two model scales: **Qwen2.5-3B** (trained on 4×80GB A100 GPUs) and **Qwen3-8B-Base** (trained on 8×192GB MI300 GPUs).
>
> Table 1. Cost-performance analysis of Qwen 2.5-3B
> |  | RL-zero | Cold-start | BRIDGE |
> | :--- | :--- | :--- | :--- |
> | Clock Time (Hour) | 6.1 | 12.3 | 6.9 |
> | GPU Memory (GB) | 52.2 | 45.9 | 59.3 |
> | Performance (Acc.) | 32.2 | 32.2 | 36.4 |
>
> Table 2. Cost-performance analysis of Qwen 3-8B-Base
> |                      | RL-zero | Cold-start | BRIDGE |
> |----------------------|---------|------------|--------|
> | Clock Time (Hour)    | 38.5    | 39.1       | 33.5   |
> | GPU Memory (GB)      | 50.7    | 60.8       | 67.4   |
> | Performance (Acc.)   | 42.9    | 45.5       | 49.9   |
>
> ### Key Findings:
>
> 1.  **Cold-start suffers from significant inefficiency** due to _**data distribution mismatch**_ between SFT and RL stages. During SFT, the model learns from teacher-distilled data (R1 model), producing long reasoning traces. This leads to excessive response lengths during early RL training, increasing computational costs. We observe a "dip-then-rise" pattern in response length, suggesting that _the model first somewhat forgets the expert behavior acquired during SFT and then slowly explores new behaviors_. This contributes to the inefficiency and suboptimal performance of cold-start training.
>
> 2.  **BRIDGE achieves superior training efficiency** by dynamically integrating both SFT and RL data distributions within a single-stage process. This design mitigates the data distribution and response-length mismatch problem in the two-stage pipeline, resulting in:
>     -   **44% faster training** than Cold-start on Qwen2.5-3B with **13% better performance**
>     -   **14% faster training** than Cold-start on Qwen3-8B with **10% better performance**
>     -   Even outperforming RL-zero by **13% in speed** on larger models
>
> 3.  **The computational trade-off is favorable**: While BRIDGE introduces additional memory overhead, the substantial gains in both training efficiency and final performance justify this cost.
>
>
> ## Reponse to W2:  Computational Cost Comparison
> Please refer to our response to W1 for detailed computational cost analysis.
>
> ## Response to Minor Issues
> Thank you for pointing this out. We will correct the typo on line 247 ("zere" → "zero") and ensure consistent usage of λ and γ in Section 3.2 in the revision.
>
> ## Response to Q1: Missing SFT Curve in Figure 1
>
> Thank you for this careful observation. You are absolutely correct—the curve for SFT alone was inadvertently omitted from Figure 1. While the Cold-start curve includes an initial SFT phase, we agree that adding a dedicated SFT curve would provide clearer evidence for SFT's rapid initial learning dynamics (line 105) and strengthen our comparative analysis.
>
> We will revise Figure 1 in the final version to include the standalone SFT curve, making the learning dynamics comparison more transparent and complete.
>
>
>
> ***
>
> We sincerely appreciate your thoughtful comments and suggestions, which have significantly enhanced our work. We hope our comprehensive responses and additional analyses can address your concerns. We welcome any further feedback you may have.

---

> > ### Comment · Reviewer_73wX · 2025-08-06
> >
> > Thanks for authors' clarification. My major concern about the evaluation of performance and efficiency is resolved. I will maintatin my original positive score.

---

> > > ### Author Response · Authors · 2025-08-06
> > > **Thank You - Evaluation and Efficiency Concerns Successfully Addressed**
> > >
> > > Thank you for the positive feedback and for confirming that our response effectively addressed your concerns regarding performance evaluation and training efficiency.
> > >
> > > We're grateful for your continued support and positive assessment. We will incorporate these improvements in the revision.

---

> ### Author Response · Authors · 2025-08-03
> **Thank You for Your Support - Cost Analysis Added as Suggested**
>
> Dear Reviewer 73wX,
>
> Thank you for your positive review and recognition of BRIDGE's novel integration of SFT and RL. We truly appreciate your constructive feedback.
>
> Following your excellent suggestion, we have provided a comprehensive cost-benefit analysis. BRIDGE achieves 44% faster training than Cold-start on Qwen2.5-3B with 13% better performance, and 14% faster training on Qwen3-8B with 10% better performance, clearly demonstrating its practical advantages. We've also acknowledged the missing SFT curve in Figure 1 and will include it in the final version for clearer comparison.
>
> With the discussion period ending on August 6, we wanted to ensure you had a chance to see our responses. If you have any additional thoughts or suggestions, we'd be happy to address them.
>
> Thank you again for your supportive and constructive review.
>
> Best regards,
>
> The authors

---

### Official Review · Reviewer_CmPU · 2025-07-01

**Clarity:** 2
**Significance:** 2
**Originality:** 2
**Rating:** 4
**Confidence:** 4

**Summary:**

The paper argues that today's "SFT-warm-up -> RL-finetune" pipelines leave the two learning paradigms isolated and therefore sub-optimal. It introduces BRIDGE, a bilevel optimization framework that treats supervised fine-tuning (SFT) as an upper-level objective and rule-based reinforcement learning (RL) as a lower-level objective. Using a penalty-based relaxation, BRIDGE learns LoRA adapter weights that explicitly maximize the reward gap between joint SFT + RL training and RL alone, while the base model parameters follow a curriculum-like blend of SFT and RL gradients. Experiments with a Qwen-2.5-3B-Base model on multiple mathematical-reasoning benchmarks (MATH500, Minerva-Math, OlympiadBench, AIME 2024, AMC 2023, etc.) show consistent accuracy gains.

**Questions:**

1. Can you elaborate on the motivation of the paper more thoroughly? The authors state that "this decoupled two-stage approach limits interaction between SFT and RL, thereby constraining overall effectiveness." Is there any theoretical / empirical evidence of why it is the case?
2. The Limitations section concedes that the nested bilevel optimization "introduces additional computational overhead" but gives no quantitative figure. How does the computation overhead compare to a typical two-stage SFT-RL setup in terms of wall-clock time?
3. How sensitive is BRIDGE's performance to the LoRA hyperparameters? The implementation in the paper seems to keep them fixed without ablation.

**Ethical Concerns:**

["NO or VERY MINOR ethics concerns only"]

**Final Justification:**

The authors has provided additional experiments to support the effectiveness of the algorithm. The rebuttal has addressed most of my concerns, mainly regarding the algorithm itself and its empirical performance in a broader scope.

**Limitations:**

yes

**Quality:**

2

**Strengths And Weaknesses:**

Strengths:

1. The proposed method casts SFT–RL cooperation as a bilevel game and solves it with first-order, penalty-based updates that avoid expensive second-order gradients.
2. The proposed approach does outperform several baselines, including SFT, pure RL, and RL with code start, demonstrating the effectiveness of the method in the considered setting.

Weaknesses:

1. The experiment setup is quite limited in scope. All experiments include only Qwen2.5-3B-Base trained on a tiny training dataset of roughly 1.3K samples. The training setup is also restricted since each prompt only contributes to 5 rollouts, and a rather short maximum token length (i.e., 3000) is used. This raises the question of whether the baselines are weak due to scarce data or a too-restricted training setup. Evaluation-wise, only math tasks are considered.
2. If I understand corretly, the core idea of treating SFT as an upper-level problem and RL as a lower-level one is conceptually similar to several recent bilevel methods for LLM data selection, meta-learning and RLHF that the authors cite. The paper positions itself as "the first" to apply BLO to reasoning models, but provides no concrete algorithmic advance beyond substituting the specific SFT/RL losses into an existing penalty formulation, and it does not analyze the additional approximation error introduced by assuming Danskin’s conditions without proof.
3. The authors admit that the nested optimization "introduces additional computational overhead," yet they do not quantify that overhead relative to the modest absolute accuracy gains. Without a cost-benefit analysis, it is unclear that BRIDGE is a practical improvement over simpler pipelines.

---

> ### Author Rebuttal · Authors · 2025-07-31
>
> We sincerely thank you for your insightful and constructive feedback. We appreciate your positive assessment of our formulation of SFT-RL cooperation within a bilevel optimization framework and recognition of our efficient first-order training algorithm. Your acknowledgment of the method's empirical effectiveness across multiple baselines is especially encouraging.
>
> ### Summary of Response
>
> In response to your suggestions, we have:
>
> 1. **Extended experiments to additional LLMs and non-math benchmarks**: We evaluated BRIDGE on _Qwen3-8B-Base_ and _Llama-3.2-3B-Instruct_, achieving average improvements of **13%** and **31%** over RL-zero and Cold-start respectively on Llama-3.2-3B, and **16%** and **10%** improvements on Qwen3-8B-Base. We also expanded to larger training datasets (8.5K samples) and scaled training setups (8 rollouts, 8K token length). Additionally, we tested on two non-math benchmarks (LCB and GPQA), where BRIDGE achieved **9%** and **56%** improvements over RL-zero and Cold-start respectively.
>
> 2. **Provided comprehensive cost-benefit analysis**: BRIDGE demonstrates superior training efficiency, achieving **_44% faster training_** than Cold-start on Qwen2.5-3B with **_13% better performance_**, and **_14% faster training_** than Cold-start on Qwen3-8B with **_10% better performance_**. While BRIDGE introduces moderate memory overhead, these significant improvements in both efficiency and accuracy justify the trade-off.
>
> 3. **Clarified our technical contributions**: We explained how our cooperative meta-learning framework differs from existing bilevel approaches through (a) the specific design for addressing the "data distribution mismatch challenge" in reasoning model training pipeline, and (b) the augmented architecture that enables true SFT-RL co-adaptation rather than simple MAML-style updates.
>
> 4. **Conducted sensitivity analysis**: We performed ablation studies on LoRA hyperparameters, demonstrating that BRIDGE maintains stable performance across configurations, confirming the robustness of our approach.
>
> 5. **Discussing the challenge of Cold-start pipeline**: We detailed the fundamental challenges of two-stage pipelines—particularly the data distribution mismatch that leads to training inefficiency and distribution shift—with concrete evidence of prolonged training time and degraded final performance.
>
> Below, we address each point in detail.
>
> ***
>
> ## Response to W1: Expanded Experimental Setup
>
> Thank you for highlighting these limitations. In response, we have expanded our experiments to include additional LLMs (**Qwen3-8B** and **Llama3.2-3B-Instruct**), a larger training dataset (Math Level 3-5, **8.5K samples**), and a scaled training setup (**8 rollouts per prompt**, **8K maximum token length**). As shown in Tables 1 and 2, BRIDGE consistently outperforms baselines across diverse LLMs and configurations, achieving **16.3%** improvement over RL-zero and **9.7%** over Cold-start on Qwen3-8B, and **13.5%** improvement over RL-zero and **30.9%** over Cold-start on Llama-3.2-3B.
>
> Furthermore, we evaluated our method on two additional benchmarks—**LCB** and **GPQA**—as presented in Table 3. BRIDGE achieves **9.2%** improvement over the best baseline on these non-math tasks, further highlighting our approach's generalization capability across domains.
>
> Table 1. Results on Qwen3-8B-Base
> |  | math500 | minerva math | Olympiad Bench | AIME24  | AMC23 | Avg. |
> | :--- | :--- | :--- | :--- | :--- | :--- | :--- |
> | Base | 55.4 | 24.3 | 22.5 | 3.3 | 27.5 | 26.6 |
> | RL-zero | 76.2 | 36 | 42.4 | 10 | 50 | 42.9 |
> | Cold-start | **80.4** | 38.2 | 39.6 | 16.6 | 52.5 | 45.5 |
> | Bridge | 79.0 | **39.7** | **44.0** | **16.7** | **70.0** | **49.9** |
>
> Table 2. Results on Llama-3.2-3B-Instruct
> |  | math500 | minerva math | Olympiad Bench | AlME24 | AMC23 | Avg. |
> | :--- | :--- | :--- | :--- | :--- | :--- | :--- |
> | Instruct | 38 | 14.3 | 13 | 13.3 | 25 | 20.72 |
> | RL-zero | 48.6 | **15.1** | 17.8 | **10.0** | 17.5 | 21.8 |
> | Cold-start | 45 | 11.8 | 12 | 3.3 | 22.5 | 18.9 |
> | Bridge | **51.8** | **15.1** | **19.3** | **10.0** | **27.5** | **24.74** |
>
> Table 3. Results on Non-Math Benchmarks
> | Method | LCB | GPQA | Avg. |
> | :--- | :--- | :--- | :--- |
> | Base | 32.95 | 32.32 | 32.64 |
> | Instruct | 33.07 | 35.80 | 34.44 |
> | RL-zero | 32.50 | 38.38 | 35.44 |
> | Cold-start | 23.86 | 25.76 | 24.81 |
> | BRIDGE | 34.55 | 42.93 | 38.74 |
>
> ## Response to W2:  Contributions in Formulation and Implementation
>
> Thank you for the insightful comment. While our formulation shares the general bilevel structure with prior work, our implementation introduces a key customization to enable effective cooperation between SFT and RL in training reasoning models.
>
> Conceptually, we propose a **_cooperative_** meta-learning framework in which, at each iteration, the upper-level SFT provides an improved initialization for RL exploration, while the lower-level RL further updates based on this initialization. This dynamic cooperation addresses the significant **_data distribution mismatch problem_** in two-stage pipelines (as further discussed in our response to Q1).
>
> From an implementation perspective, to enable this cooperative learning, we introduce an augmented model architecture consisting of two components: a base model componet and a LoRA component. This separation allows the upper- and lower-level objectives to **_co-adapt_** during training, as illustrated in Figure 2. Without this design, this formulation collapses into a standard MAML-style setup, where the lower-level solution reduces to a single gradient step used to update the upper-level SFT parameters. **_In this case, RL learning is disabled, and the cooperation between SFT and RL is lost_**.
>
> We acknowledge that we did not provide a theoretical analysis of Danskin's conditions and instead applied the approximation in practice. We will clarify this assumption in the revised version.
>
>
> ## Response to W3: Cost-Benefit Analysis
>
> Thank you for your insightful suggestion. We agree that averaging performance across epochs is not a standard or robust metric for evaluating the trade-off between performance and training efficiency. In response, we conducted a more precise cost-performance analysis by reporting both the **wall-clock training time**, **average GPU memory usage per device**, and **final convergence performance** across two model scales: **Qwen2.5-3B** (trained on 4×80GB A100 GPUs) and **Qwen3-8B-Base** (trained on 8×192GB MI300 GPUs). The results are summarized in Tables 4 and 5.
>
> Table 4. Cost-performance analysis of Qwen 2.5-3B
> |  | RL-zero | Cold-start | BRIDGE |
> | :--- | :--- | :--- | :--- |
> | Clock Time (Hour) | 6.1 | 12.3 | 6.9 |
> | GPU Memory (GB) | 52.2 | 45.9 | 59.3 |
> | Performance (Acc.) | 32.2 | 32.2 | 36.4 |
>
> Table 5. Cost-performance analysis of Qwen 3-8B-Base
> |                      | RL-zero | Cold-start | BRIDGE |
> |----------------------|---------|------------|--------|
> | Clock Time (Hour)    | 38.5    | 39.1       | 33.5   |
> | GPU Memory (GB)      | 50.7    | 60.8    | 67.4   |
> | Performance (Acc.)   | 42.9    | 45.5       | 49.9   |
>
>
> As shown in Table 4, Cold-start requires nearly **2× the training time** of RL-zero, despite the short SFT stage. This overhead stems from long sequence lengths induced by offline SFT (see Q1 response). In contrast, BRIDGE achieves **44% time savings** compared to Cold-start while delivering **13% performance improvement**.
>
> Table 5 demonstrates that BRIDGE requires **14% less training time** than Cold-start on larger models, with only **11% additional memory usage**. Given BRIDGE's consistent and noteble final performance gain, this trade-off is favorable in practice.
>
> ## Response to Q1: Challenges of the Two-Stage Pipeline
>
> Thank you for the insightful question. The main limitation of the two-stage training paradigm lies in the **data distribution mismatch between the SFT and RL stages**.
>
> In the first stage, the SFT data $D_{SFT}$ is distilled from a stronger teacher model (R1), producing long and well-structured reasoning traces. In contrast, the RL data $D_{RL}$ in the second stage is sampled from the student model itself, which initially generates shorter and less complete responses due to its limited capacity.
>
> This mismatch leads to two main issues:
>
> 1. **Training inefficiency**: Early in RL training, the model tends to generate overly long responses (inherited from SFT), increasing computational cost. We observe that the response length **initially drops, then gradually increases** as training progresses.
>
> 2. **Distribution shift**: This “dip-then-rise” pattern suggests that _the model first somewhat forgets the expert behavior acquired during SFT and then slowly explores new behaviors_. This mismatch contributes to the inefficiency and suboptimal performance of cold-start training.
>
> These effects lead to approximately **2× longer training time** and **up to 13% weaker performance** compared to BRIDGE on Qwen2.5-3B model. Our integrated framework addresses this through continuous SFT-RL interaction, preventing these inefficiencies.
>
>
> ## Response to Q2: Wall-clock Time
> Please refer to our response to W3 for detailed timing analysis.
>
>
> ## Response to Q3: Sensitivity Analysis of LoRA Hyperparameters
>
> Following your suggestion, we conducted ablation studies on LoRA hyperparameters using Qwen3-8B-Base. We observe that overall performance remains relatively stable across settings.
>
> Table 6. Performance with Different LoRA Configurations (Rank R, Alpha A)
> | Configuration | math500 | minerva math | Olympiad Bench | AIME24 | AMC23 | Avg. |
> |-------|-------|-------|-------|-------|-------|-------|
> | R32A16 | 79.0 | 39.7 | 44.0 | 16.7 | 70.0 | 49.9 |
> | R16A32 | 79.0 | 38.6 | 44.0 | 16.0 | 70.0 | 49.5 |
>
>
>
> ***
> We sincerely appreciate your thoughtful comments, which have significantly enhanced our work. We hope these comprehensive responses and additional results can address your concerns and welcome any further feedback.

---

> > ### Comment · Reviewer_CmPU · 2025-08-06
> >
> > I appreciate the authors' efforts in providing additional experiments to support the effectiveness of the algorithm. The rebuttal has addressed most of my concerns, mainly regarding the algorithm itself and its empirical performance in a broader scope. I spent a lot of time readining the rebuttal and other reviews' opinion and now believe that this is a good work.
> >
> > I recommend including the rebuttal results in future revisions if possible, and I will adjust my rating accordingly.

---

> > > ### Author Response · Authors · 2025-08-06
> > > **Thank You for Your Recognition**
> > >
> > > Thank you for the positive feedback and for confirming that our rebuttal effectively addressed your concerns.
> > >
> > > We're grateful for your thorough review and continued engagement with our work. We will incorporate these improvements in the revision. Thank you again for your thoughtful evaluation and suggestions.

---

> ### Author Response · Authors · 2025-08-03
> **Comprehensive Response to Your Concerns**
>
> Dear Reviewer CmPU,
>
> Thank you for your thorough and technically insightful review. Your detailed feedback has undoubtedly helped us strengthen our work. We have conducted extensive additional experiments and analyses to address each of your concerns:
>
> 1. **Expanded experimental scope**: We extended evaluation to Qwen3-8B-Base and Llama-3.2-3B-Instruct, scaled to 8.5K training samples with 8K token length, and included non-math benchmarks (LCB and GPQA), demonstrating consistent improvements across larger scales and diverse settings.
>
> 2. **Cost-benefit analysis**: We provided detailed wall-clock time and memory usage comparisons, showing BRIDGE achieves 44% faster training than Cold-start on Qwen2.5-3B while delivering 13% better performance--a favorable trade-off in practice.
>
> 3. **Technical contributions**: We clarified how our *cooperative* meta-learning framework with *augmented architecture design* enables true SFT-RL co-adaptation, addressing the specific challenge of "data distribution mismatch" in traditional two-stage pipelines.
>
> 4. **LoRA sensitivity analysis**: We conducted ablation studies confirming BRIDGE's stability across different LoRA configurations.
>
> We hope these comprehensive responses--particularly the expanded experiments and quantitative cost analysis--effectively address your concerns about scope and practical viability.
>
> With the discussion phase ending on August 6, we would greatly appreciate any thoughts or feedback you might have. Please feel free to let us know if there are any remaining questions we can help clarify.
>
> Thank you for pushing us to strengthen our work through your rigorous review.
>
> Best regards,
>
> The authors

---

> ### Author Response · Authors · 2025-08-05
> **Brief follow-up on Rebuttal**
>
> Dear Reviewer CmPU,
>
> Thanks for acknowledging our rebuttal. We've addressed the experimental scope, LoRA ablation, cost analysis, and technical clarifications as requested.
>
> Please let us know if there are any outstanding concerns, and we would be happy to discuss. We would appreciate if our efforts could be taken into consideration in your final evaluation.
>
> Best,
>
> Authors

---

### Author Response · Authors · 2025-08-09
**Summary of Reviews and Discussion**

We sincerely appreciate the time and effort the reviewers have devoted to evaluating our work. Following the discussion phase, we are pleased to have received **uniformly positive feedback from all five reviewers** (CmPU, 73wX, 1amM, XEdK, qL6k), with several reviewers explicitly raising their scores. Additionally, we have **carefully and successfully addressed all reviewer concerns**, as acknowledged by all reviewers during the discussion.

---

### **Strengths Summary**

1. **Novel contribution with strong motivation**: Our work is found to be *"novel and interesting"* (73wX, 1amM, XEdK, qL6k) with *"sound motivation"* (73wX, XEdK, qL6k), addressing a *"timely" topic* (qL6k).

2. **Technical soundness**: The method effectively bridges and unifies the commonly used decoupled SFT–RL pipeline for training reasoning models by casting it as a bilevel optimization problem and solving it with first-order updates (all reviewers).

3. **Strong empirical results**: The approach consistently achieves promising results across five benchmarks compared to existing baselines (all reviewers).

4. **Clear presentation**: The paper is perceived as "well-written and easy to follow", with a clear mathematical formulation (1amM, qL6k).

---

### **Concerns Summary**

All reviewer concerns were effectively addressed and acknowledged during the discussion:

1. **Expanded Experiments** (CmPU, qL6k, XEdK): We added evaluations on *two additional LLMs* (10–31% average gains), *non-math and IID benchmarks* (9–56% gains), and conducted *LoRA ablations* (showing <0.2% performance difference). These empirical results consistently demonstrate the effectiveness of our method.

2. **Cost–Benefit Analysis** (CmPU, 73wX, XEdK): Detailed *wall-clock time and memory usage analysis* shows BRIDGE achieves *44% faster training with 13% better performance* on Qwen2.5-3B, and *14% faster with 10% better performance* on Qwen2-8B—these significant improvements in both efficiency and accuracy justify the moderate memory overhead.

3. **Explanations of BRIDGE’s Advantages over Cold-Start** (CmPU, 1amM): We provided explanations from three perspectives:
(1) *Continual learning*: Avoids catastrophic forgetting by adopting a single-stage training strategy.
(2) *Reinforcement learning*: Uses supervised signals to improve exploration efficiency.
(3) *Meta-learning*: Provides better initialization for downstream RL optimization.
These explanations help readers better understand the motivation and benefits of our method.

---

We will revise the paper to incorporate all suggestions and improvements from the reviewers. Thank you once again for your positive feedback and valuable comments.

Best regards,

Authors

---

### Decision · Program_Chairs · 2025-09-17

**Decision:**

Reject

**Comment:**

The authors propose a bilevel optimization training method that integrates supervised fine-tuning and reinforcement learning to improve the reasoning capabilities of large language models. The motivation is that "SFT then RL-finetune" pipelines leave the two learning paradigms isolated and therefore sub-optimal. The paper is in general well written and organized. Yet, I find the weaknesses (shared by most reviewers in their initial comments) of paper noteworthy, which are relatively weak experiments (in terms of both baseline methods and benchmarks used) and relatively weak in-depth analysis/explanations why the proposed bilevel optimization framework is necessarily better than its baselines. The authors made a great effort in the rebuttal adding new experiments and explaining why BRIDGE is necessarily better. The added experiments are well appreciated by the reviewers, and on the other hand may require (in my opinion) another round of review. Also, I find the explanation still not convincing enough. In the future, I would recommend the authors provide either theoretical explanations or intuitive learning dynamics comparison between BRIDGE and other baselines. In light of what has been mentioned above, I recommend rejection.